# BEYOND TWO-STAGE TRAINING: COOPERATIVE SFT AND RL FOR LLM REASONING

## ABSTRACT

Reinforcement learning (RL) has proven effective in incentivizing the reasoning abilities of large language models (LLMs), but suffers from severe efficiency challenges due to its trial-and-error nature. While the common practice employs supervised fine-tuning (SFT) as RL warmup, its distribution mismatches the policy's rollouts. This mismatch produces a dip-then-rise dynamic: early RL forgets SFT-acquired behavior and slowly re-explores, resulting in limited effectiveness and inefficient exploration. We introduce BRIDGE, a novel method to employ *bilevel optimization* to facilitate better cooperation between these training paradigms. By conditioning the SFT objective on the optimal RL policy, our approach enables SFT to meta-learn how to guide RL's optimization process. During training, the lower-level performs RL updates while simultaneously receiving SFT supervision, while the upper-level explicitly maximizes the *cooperative gain*—the performance advantage of joint SFT-RL training over RL alone. Empirical evaluations across three LLMs and five reasoning benchmarks demonstrate that our method consistently outperforms baselines and achieves a better balance between effectiveness and efficiency. Specifically, BRIDGE achieves 44% faster training with a 13% performance gain on Qwen2.5-3B, and 14% faster training with a 10% improvement on Qwen3-8B.

## 1 INTRODUCTION

The emergence of OpenAI's o1 (OpenAI) and DeepSeek-R1 (DeepSeek-AI et al., 2025) marks a significant advance in LLM reasoning capabilities, particularly for challenging tasks such as mathematics (Cobbe et al., 2021; Hendrycks et al., 2021b) and programming (Chen et al., 2021; Codeforces, 2025). The key technique driving this progress is large-scale, rule-based RL. However, the inherently trial-and-error nature of RL renders the training process highly inefficient. An alternative approach is SFT on curated long chain-of-thought (CoT) datasets, which enables models to rapidly acquire effective reasoning patterns through imitation learning. While more sample-efficient, SFT typically exhibits poorer performance and generalization than RL (Chu et al., 2025).

In practice, production-scale pipelines often use a two-stage paradigm, warming up with SFT before applying RL. This decoupled design induces a distribution mismatch between teacher-distilled SFT traces and the student's on-policy rollouts. Empirically, it yields a dip-then-rise dynamic, indicating that it suffers from catastrophic forgetting; and because the RL stage lacks sustained, policy-aware guidance, exploration is inefficient. Consequently, the two stages fail to synergize and the pipeline fails to leverage the complementary strengths of imitation and reinforcement. This raises a natural question:

> *Can we design a training framework that enables meaningful synergy between SFT and RL, ensuring their cooperation yields performance superior to standalone RL?*

To explore this possibility, we first propose a simple baseline that alternates between SFT and RL updates during training. Despite its simplicity, this approach improves both convergence efficiency and final performance. However, such *independent updates* cannot guarantee improvements over RL alone, as not all SFT updates benefit RL optimization. Building on this insight, we develop BRIDGE, a cooperative learning framework based on bilevel optimization, where SFT serves as the upper-level problem and RL as the lower-level problem. By solving this nested structure—with the SFT objective

explicitly conditioned on the RL solution—SFT provides *targeted* guidance that directly supports RL's optimization process.

Specifically, BRIDGE employs an augmented model architecture comprising two learnable components: a base model and a LoRA module. The base model is optimized through the lower-level RL objective, while the LoRA parameters are updated via the upper-level supervised objective. To solve this bilevel problem, we adopt a first-order, penalty-based relaxation method. The relaxed lower-level update blends SFT and RL gradients, while the upper-level update ***explicitly maximizes the cooperative gain—the performance advantage of joint SFT-RL training over RL-only optimization***. In this way, the lower level realizes the cooperation between two objectives, while the upper level ensures this cooperation yields superior performance.

To validate the effectiveness of our approach, we conduct experiments with three LLMs across five diverse benchmark datasets covering both standard and competition-level math reasoning tasks. Results demonstrate that BRIDGE consistently outperforms all baselines—including SFT, RL-zero, cold-start, and our naive alternating baseline—while requiring less wall-clock training time. These improvements confirm the benefits of tightly coupling SFT and RL through bilevel optimization rather than treating them as separate phases.

Our work makes the following contributions:

1. **Comparative analysis of reasoning training paradigms.** We systematically analyze three prevalent strategies for training large reasoning models. Our analysis reveals that the lack of interaction in two-stage pipelines prevents SFT and RL from effectively synergizing and leads to catastrophic forgetting and inefficient exploration. To mitigate these issues, we introduce a simple alternating baseline that achieves superior performance.

2. **A bilevel optimization framework for integrating SFT and RL.** To achieve deeper cooperation between SFT and RL, we propose BRIDGE, a bilevel optimization method that formalizes SFT as the upper-level and RL as the lower-level problem. Built on an augmented model architecture and solved using penalty-based relaxation, BRIDGE explicitly maximizes the cooperative gain—ensuring joint training outperforms standalone RL.

3. **Empirical validation on mathematical reasoning benchmarks.** We conduct extensive experiments with three LLMs across five mathematical reasoning benchmarks. BRIDGE consistently outperforms five baselines in both accuracy and training efficiency, demonstrating the practical benefits of tightly integrated SFT-RL optimization.

## 2 PRELIMINARIES

We begin by reviewing three prevalent fine-tuning strategies for training reasoning models, conduct a comparative analysis, and discuss limitations of the popular two-stage method. We then introduce a simple yet effective baseline that improves upon it.

### 2.1 FINE-TUNING METHODS FOR REASONING MODELS

We consider a language model parameterized by $\boldsymbol{\theta}$, which defines a conditional distribution $\pi(y|x; \boldsymbol{\theta})$ over output sequences $y$ given input sequences $x$. This work focuses on three widely used methodologies for fine-tuning $\boldsymbol{\theta}$ to enhance the model's reasoning capabilities.

**Supervised Fine-Tuning.** In supervised fine-tuning, we assume access to a curated dataset $\mathcal{D}_{\text{SFT}} := \{(x, r, y)\}$ consisting of input prompts $x$, intermediate reasoning steps $r$ distilled from larger reasoning models or annotated by human experts, and final answers $y$. The training objective maximizes the log-likelihood of generating both the reasoning process and the final answer:

$$\max_{\boldsymbol{\theta}} \ J_{\text{SFT}}(\boldsymbol{\theta}) := \mathbb{E}_{(x,r,y)\sim\mathcal{D}_{\text{SFT}}} \left[ \log \pi\left(r, y \mid x; \boldsymbol{\theta}\right) \right]. \tag{1}$$

This approach encourages the model to not only produce correct answers but also to imitate expert reasoning steps that lead to those answers.

**Reinforcement learning with verifiable rewards (RLVR).** RLVR has gained increasing attention for its effectiveness in training advanced reasoning models such as DeepSeek-R1 (DeepSeek-AI et al.,

2025). Given a dataset $\mathcal{D}_{\mathrm{RL}} := \{(x, y)\}$ with verifiable outputs—such as mathematics competition problems—the objective of rule-based RL is formulated as:

$$\max_{\boldsymbol{\theta}} \ J_{\mathrm{RL}}(\boldsymbol{\theta}) := \mathbb{E}_{(x,y)\sim\mathcal{D}_{\mathrm{RL}},\ (\hat{r},\hat{y})\sim\pi(\cdot|x;\boldsymbol{\theta})} \left[R(\hat{y}, y)\right]$$
$$- \mathbb{E}_{(x,y)\sim\mathcal{D}_{\mathrm{RL}}} \left[D_{\mathrm{KL}} \left(\pi(\cdot \mid x; \boldsymbol{\theta}) \,\|\, \pi_{\mathrm{ref}}(\cdot \mid x)\right)\right] \tag{2}$$

where $\pi_{\mathrm{ref}}$ is a fixed reference model and $R(\hat{y}, y)$ is a *rule-based reward function* that evaluates prediction correctness using a binary signal:

$$R(\hat{y}, y) = \begin{cases} 1, & \text{if } \hat{y} \equiv y, \\ 0, & \text{otherwise} \end{cases} \tag{3}$$

Here, $y$ denotes the ground-truth answer and $\hat{y}$ is the model's predicted output. The equivalence relation $\hat{y} \equiv y$ is typically computed by a rule-based verifier. This objective is commonly solved using policy optimization methods such as Proximal Policy Optimization (PPO) (Schulman et al., 2017) or Group Relative Policy Optimization (GRPO) (DeepSeek-AI et al., 2025).

**Two-Stage Cold Start.** In practice, the common recipe uses SFT as a warm-up stage before applying RL. This two-stage approach, often referred to as "cold start," ensures that the model first learns to imitate expert reasoning patterns, providing a strong initialization for subsequent RL training.

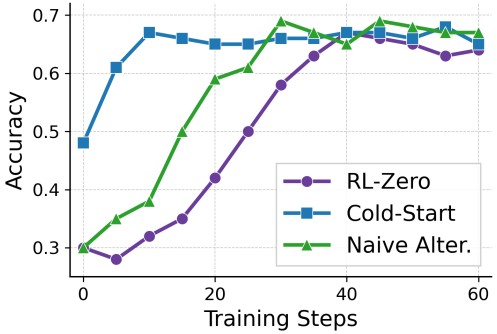

Figure 1: Comparison of Training Methods.

**Algorithm 1:** A Simple Alternating Method
1: Initialize parameters $\theta_0$; datasets $D_{\mathrm{SFT}}$, $D_{\mathrm{RL}}$; learning rates $\alpha_{\mathrm{SFT}}$, $\alpha_{\mathrm{RL}}$; total steps $T$
2: **for** $t = 1$ to $T$ **do**
3:     // RL step
4:     Sample $(x_t, y_t) \sim D_{\mathrm{RL}}$
5:     Generate solution with $\pi_{\theta_{t-1}}(x_t)$
6:     Compute RL objective $J_{\mathrm{RL}}$ using (2)
7:     $\theta'_{t-1} \leftarrow \theta_{t-1} + \alpha_{\mathrm{RL}} \nabla J_{\mathrm{RL}}(\theta_{t-1})$
8:     // SFT step
9:     Sample example $(x_t, r_t, y_t) \sim D_{\mathrm{SFT}}$
10:    Compute SFT objective $J_{\mathrm{SFT}}$ using (1)
11:    $\theta_{t-1} \leftarrow \theta'_{t-1} + \alpha_{\mathrm{SFT}} \nabla J_{\mathrm{SFT}}(\theta'_{t-1})$
12: **end for**

## 2.2 COMPARISON OF FINE-TUNING METHODS

We evaluate these methods on mathematics problems at the grade 3–5 level. Figure 1 illustrates the evolution of test accuracy during training. We observe that *while SFT provides effective initialization and rapid early convergence for cold-start training, it contributes little to final convergence performance*. This results in faster initial accuracy improvements, but performance plateaus with minimal gains in the later stages of the two-phase pipeline. In contrast, RL alone converges more slowly but eventually achieves comparable final performance.

These results suggest that SFT and RL offer complementary strengths in reasoning tasks: SFT facilitates rapid initial learning, while RL enables better asymptotic performance. However, the naïve two-stage combination in cold-start training fails to fully exploit these complementary advantages. We identify two key limitations:

1. **Catastrophic forgetting:** The two-stage paradigm suffers from catastrophic forgetting—the model loses valuable SFT-acquired knowledge when transitioning to RL training. This phenomenon is evident in the response length dynamics during cold-start's second stage (see the length dynamics in Figure 3). Response lengths initially drop sharply before gradually recovering, exhibiting a "dip-then-rise" pattern that indicates the model first forgets some expert behaviors before slowly exploring new strategies.
2. **Inefficient exploration:** Despite effective SFT initialization, online RL frequently encounters inefficient exploration, particularly on challenging problems where LLMs fail to generate reward-yielding solutions. LLMs often become trapped in local optima, unable to make further progress (see the reward dynamics in Figure 3). Moreover, once the initial SFT phase concludes, it cannot provide continued guidance for difficult problems.

These limitations motivate integrating SFT and RL training within a unified framework.

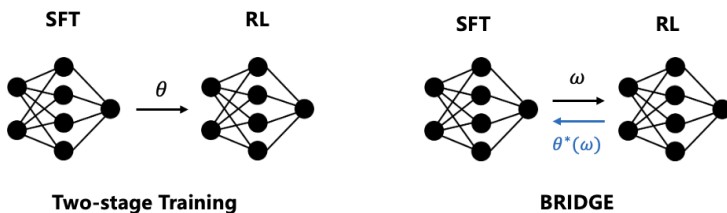

Figure 2: Comparison of two training methods.

## 2.3 A SIMPLE ALTERNATING BASELINE

To investigate the potential synergy between two methods, we design a simple alternating optimization strategy, as outlined in Algorithm 1. This approach alternates between RL steps, which explore novel reasoning strategies, and SFT steps, which imitate expert reasoning patterns.

As shown in Figure 1, this alternating strategy converges faster than pure RL and achieves better final performance than both standalone SFT and two-stage cold-start training. While this integration yields empirical gains, the current formulation treats SFT and RL as **independent update** processes with **no guarantee** that alternating updates will consistently outperform RL method alone. This limitation prompts us to investigate: *Can we develop a principled optimization framework where SFT and RL updates are inherently coordinated to guarantee improvements over pure RL?*

## 3 METHODOLOGY

In this section, we propose BRIDGE, a framework that tightly couples SFT and RL through a cooperative meta-learning approach. We first introduce the mathematical formulation, then present the learning algorithm and explanations.

### 3.1 BRIDGE: COOPERATIVE META-LEARNING FOR SFT AND RL

Given an SFT dataset $\mathcal{D}_{\mathrm{SFT}}$ and an RL dataset $\mathcal{D}_{\mathrm{RL}}$ (defined in Section 2.1), our objective is to integrate policy optimization (Eq. equation 2) with supervised learning (Eq. equation 1). We propose the following cooperative meta-learning formulation:

$$\max_w \quad J_{\mathrm{SFT}}(\theta^*(w), w) := \mathbb{E}_{(x,r,y) \sim \mathcal{D}_{\mathrm{SFT}}} \left[ \log \pi \left( r, y \mid x;\ \theta^*(w), w \right) \right]$$

$$\text{s.t.} \quad \theta^*(w) := \arg\max_\theta \left\{ \mathbb{E}_{(x,y) \sim \mathcal{D}_{\mathrm{RL}},\ (\hat{r},\hat{y}) \sim \pi(\cdot | x; \theta, w)} \left[ R(\hat{y}, y) \right] \right.$$

$$\left. - \mathbb{E}_{(x,y) \sim \mathcal{D}_{\mathrm{RL}}} \left[ D_{\mathrm{KL}} \left( \pi(\cdot \mid x; \theta, w) \,\|\, \pi_{\mathrm{ref}}(\cdot \mid x) \right) \right] \right\}. \tag{4}$$

where $\theta$ denotes the base model parameters and $w$ represents the Low-Rank Adaptation (LoRA) weights (Hu et al., 2021). Together, they form an augmented model with parameters $\bar{\theta} := [\theta, w]$.

For clarity, we express Equation equation 4 in simplified notation:

$$\max_w \quad J_{\mathrm{SFT}}(w, \theta^*(w)),$$
$$\text{s.t.} \quad \theta^*(w) := \arg\max_\theta J_{\mathrm{RL}}(\theta, w). \tag{5}$$

This formulation exhibits a bilevel optimization structure inspired by the leader-follower game. SFT acts as the leader (teacher) with access to the RL follower's (student's) optimal response $\theta^*(w)$, enabling it to provide targeted guidance. Conversely, RL optimizes the base parameters $\theta$ given the auxiliary support from SFT through $w$. During training, these components interact dynamically, resulting in better cooperation. As illustrated in Figure 2, this structure enables *bidirectional* information flow—where RL's optimal solution becomes visible to SFT—in contrast to the *unidirectional* flow of traditional two-stage approaches.

From a *meta-learning* perspective, BRIDGE implements cooperative framework where, at each iteration, the upper-level SFT provides an improved initialization for RL exploration, while the lower-level RL refines this initialization through reward-based optimization. This framework *adaptively*

*extracts the most beneficial information from SFT to enhance RL training*, as SFT guidance may not always be uniformly beneficial.

The single-stage cooperative meta-learning design provides three advantages: (1) it avoids catastrophic forgetting of the two-stage pipeline through unified single-stage training; (2) it improves exploration efficiency via continuous supervised guidance; and (3) it guarantees RL performance gains by enabling SFT to meta-learn how to guide RL, strategically transferring beneficial knowledge.

**Architectural Design Rationale.** The augmented model architecture, comprising base model parameters $\theta$ and LoRA parameters $w$, is essential for enabling cooperative learning. This separation allows the upper- and lower-level objectives to *co-adapt* during training, as illustrated in Figure 2. Without this architectural separation, our formulation (Equation equation 4) would collapse to a Model-Agnostic Meta-Learning (MAML)-style setup (Finn et al., 2017), where the lower-level solution reduces to a single gradient step used to update the upper-level SFT parameters. In this case, RL learning is disabled, and the cooperation between SFT and RL is lost.

## 3.2 LEARNING ALGORITHM

To solve the bilevel optimization problem in Eq. equation 5, we employ penalty-based methods (Shen & Chen, 2023; Shen et al., 2025) to avoid expensive second-order derivative computations. We first reformulate equation 5 as a single-level problem amenable to efficient first-order optimization.

We define the penalty function measuring the sub-optimality of the lower-level problem as:

$$p(w, \theta) = \max_{\theta'} J_{\text{RL}}(\theta', w) - J_{\text{RL}}(\theta, w). \tag{6}$$

This penalty quantifies the optimality gap: $p(w, \theta) = 0$ if and only if $\theta$ maximizes $J_{\text{RL}}(\cdot, w)$.

Given a penalty weight $\lambda \in (0, 1)$, we obtain the penalized reformulation:

$$\max_{\theta, w} \mathcal{L}(\theta, w) := (1 - \lambda) J_{\text{SFT}}(\theta, w) - \lambda p(w, \theta). \tag{7}$$

The penalty weight $\lambda$ follows an annealing schedule: starting from a small value to warm-start training on supervised data, then gradually increasing to enforce the bilevel constraint more strictly.

Since $\max_{\theta'} J_{\text{RL}}(\theta', w)$ depends only on $w$, the gradient with respect to $\theta$ simplifies to:

$$\theta^{k+1} = \theta^k + \alpha \left[ (1 - \lambda) \nabla_\theta J_{\text{SFT}}(\theta, w) + \lambda \nabla_\theta J_{\text{RL}}(\theta, w) \right] \tag{8}$$

For the gradient with respect to $w$, we invoke Danskin's theorem. Assuming $J_{\text{RL}}(\cdot, w)$ satisfies the required regularity conditions, we have:

$$\nabla_w \max_{\theta'} J_{\text{RL}}(\theta', w) = \nabla_w J_{\text{RL}}(\theta^*(w), w), \tag{9}$$

where $\theta^*(w) = \arg\max_\theta J_{\text{RL}}(\theta, w)$. In practice, we approximate $\theta^*(w)$ by taking a single gradient ascent step with respect to the RL objective:

$$\hat{\theta} = \theta + \alpha \nabla_\theta J_{\text{RL}}(\theta, w), \tag{10}$$

yielding the approximate gradient update for $w$:

$$\nabla_w \mathcal{L}(\theta, w) \approx (1 - \lambda) \nabla_w J_{\text{SFT}}(\theta, w) + \lambda \left[ \nabla_w J_{\text{RL}}(\theta, w) - \nabla_w J_{\text{RL}}(\hat{\theta}, w) \right]. \tag{11}$$

The overall algorithm of BRIDGE is presented in Algorithm 2.

## 3.3 INTUITION BEHIND THE UPDATE RULES

**Lower-level update: Curriculum-weighted gradient fusion.** The update rule for $\theta$ in Eq. equation 8 performs a convex combination of SFT and RL gradients. As $\lambda$ increases from 0 to 1 during training, the algorithm smoothly transitions from pure imitation learning to pure reinforcement learning.

This adaptive curriculum (Bengio et al., 2009) reflects the model's evolving capabilities: early in training, when the base model lacks strong reasoning abilities, it benefits primarily from imitating

---

**Algorithm 2:** Learning Algorithm of BRIDGE

---

1: Initialize augmented parameters $\bar{\theta}^0 = (\theta^0, w^0)$, and auxiliary parameters $\hat{\theta}^0 := \theta^0$; learning rates $\alpha, \beta$; penalty weight $\lambda$; number of iterations $K$

2: **for** $k = 0$ to $K - 1$ **do**

3:     Sample mini-batches $\mathcal{B}_{\text{SFT}} \sim \mathcal{D}_{\text{SFT}}$ and $\mathcal{B}_{\text{RL}} \sim \mathcal{D}_{\text{RL}}$

4:     // Compute base objectives

5:     Compute $J_{\text{SFT}}(\theta^k, w^k)$, $J_{\text{RL}}(\theta^k, w^k)$ and $J_{\text{RL}}(\hat{\theta}^k, w^k)$ on $\mathcal{B}_{\text{SFT}}$ and $\mathcal{B}_{\text{RL}}$

7:     // Define composite objectives

8:     $J_{\text{Joint}}(\theta^k, w^k) = (1 - \lambda)J_{\text{SFT}}(\theta^k, w^k) + \lambda J_{\text{RL}}(\theta^k, w^k)$

9:     $J_{\text{Gain}}(w^k) = (1 - \lambda)J_{\text{SFT}}(\theta^k, w^k) + \lambda[J_{\text{RL}}(\theta^k, w^k) - J_{\text{RL}}(\hat{\theta}^k, w^k)]$

10:     // Update base parameters via joint objective

11:     $\theta^{k+1} \leftarrow \theta^k + \alpha\nabla_\theta J_{\text{Joint}}(\theta^k, w^k)$

12:     // Update auxiliary parameters via pure RL

13:     $\hat{\theta}^{k+1} \leftarrow \hat{\theta}^k + \alpha\nabla_{\hat{\theta}} J_{\text{RL}}(\hat{\theta}^k, w^k)$

14:     // Update LoRA parameters to maximize cooperative gain

15:     $w^{k+1} \leftarrow w^k + \beta\nabla_w J_{\text{Gain}}(w^k)$

16: **end for**

---

expert demonstrations. As the model develops competence in generating correct solutions, it can increasingly leverage reward signals through exploration, making RL updates progressively more valuable.

**Upper-level update: Maximizing cooperative gain.** The update for $w$ in Eq. equation 11 solves the bilevel problem by finding LoRA parameters $w$ that ensure the RL-optimized model $\theta^*(w)$ also excels on the supervised dataset $\mathcal{D}_{\text{SFT}}$.

The update in Eq. equation 11 can be interpreted as performing gradient ascent on the following objective:

$$f(\theta, w) = (1 - \lambda)\underbrace{J_{\text{SFT}}(\theta, w)}_{\uparrow \text{ likelihood on expert data}} + \lambda\underbrace{\left[J_{\text{RL}}(\theta, w) - J_{\text{RL}}(\hat{\theta}, w)\right]}_{\uparrow \text{ cooperative gain: SFT-RL vs RL-only}} \tag{12}$$

The first term maintains alignment with expert reasoning patterns, while the second term—**the cooperative advantage**—quantifies how much the joint SFT-RL optimization (using $\theta$) outperforms pure RL training (using $\hat{\theta}$). By maximizing this advantage term, the algorithm *explicitly encourages cooperation between supervised and reinforcement learning, ensuring their combination yields superior performance compared to RL alone.*

## 4 EXPERIMENT

### 4.1 SETTINGS

**Datasets.** We use LIMR (Li et al., 2025) (1.3k problems; Qwen2.5-3B) and MATH (Hendrycks et al., 2021a) (8.5k; lama-3.2-3B-Instruct and Qwen3-8B) for RL training For the SFT dataset, we pair queries from LIMR and MATH with corresponding intermediate reasoning traces extracted from DeepMath-103k (He et al., 2025), which were distilled from the DeepSeek-R1 model. We evaluate on five mathematical reasoning benchmarks: MATH500 (Hendrycks et al., 2021a), Minerva Math (Lewkowycz et al., 2022), OlympiadBench (He et al., 2024), AIME 2024, and AMC 2023, and two out-of-distribution (OOD) benchmarks (LiveCodeBench v5 (Jain et al., 2024) and GPQA (Rein et al., 2024)) to assess generalization.

**Models.** To demonstrate the generality of our approach, we experiment with three LLMs: Qwen2.5-3B (Yang et al., 2024), Llama-3.2-3B-Instruct (Grattafiori et al., 2024), and Qwen3-8B-Base (Yang et al., 2025). All models use prompt formats consistent with SimpleRL (Zeng et al., 2025).

**Implementation Details.** We use Verl (Sheng et al., 2024) for RL training; full settings are in Appendix B.

## 4.2 BASELINES

We compare BRIDGE against five baselines on the same base models:

**Base/Instruct Model.** Base or instruction-tuned model with no reasoning-specific training.

**Supervised Fine-Tuning (SFT).** Trained only on curated reasoning traces by SFT (no RL).

**RL-Zero.** RL applied to the base model from scratch (no SFT warm-up).

**Cold-Start** Two-stage SFT then RL, with fully decoupled objectives.

**Naive Alternating.** Alternates SFT and RL updates without a cooperative objective.

## 4.3 EXPERIMENTAL RESULTS

| Method | MATH 500 | Minerva Math | Olympiad Bench | AIME24 | AMC23 | Average |
|--------|----------|--------------|----------------|--------|-------|---------|
| Base | 32.4 | 11.8 | 7.9 | 0.0 | 20.0 | 14.4 |
| SFT | 53.4 | 18.8 | 21.5 | 3.3 | 42.5 | 27.9 |
| RL-zero | 64.4 | 26.5 | 27.0 | 3.3 | 40.0 | 32.2 |
| Cold-start | 66.0 | 24.3 | 26.8 | 9.0 | 35.0 | 32.2 |
| Naive Alter. | 65.2 | 25.3 | 27.1 | 6.7 | 42.5 | 33.4 (+3.7%) |
| BRIDGE | 66.2 | 23.9 | 28.9 | 13.3 | 47.5 | 36.0 (+11.8%) |

Table 1: Performance on Qwen2.5- 3B. Average gains over Cold-start are highlighted in blue.

**Main results across three LLMs.** Across five math benchmarks and three LLMs (Tables 1, 2, 3), BRIDGE attains the highest average accuracy. Cold-start behaves inconsistently across backbones: on Llama-3.2-3B-Instruct it is lower than RL-zero on average, whereas on Qwen3-8B-Base it exceeds RL-zero; this suggests that the two stage SFT then RL pipeline can constrain subsequent exploration and is not reliably optimal. Naive Alternating exceeds RL-zero and Cold-start on Qwen2.5-3B and Qwen3-8B-Base, indicating that training SFT and RL at the same time helps, although the gains are limited. In contrast, BRIDGE consistently outperforms all methods on all three backbones, with average improvements over Cold-start of 11.8% on Qwen2.5-3B, 30.9% on Llama-3.2-3B-Instruct, and 9.7% on Qwen3-8B-Base. Unlike naive alternating, BRIDGE treats SFT and RL as a cooperative objective and adapts the supervision so that it benefits RL objective; this coupling yields robust and transferable gains across instruction tuned and base models and remains effective at larger scales.

**Generalization to More Challenging Math Tasks** Baseline methods tend to yield larger improvements on relatively easier benchmarks but generalize poorly to more complex reasoning tasks. For example, the Cold-start underperforms RL-zero on Minerva Math, OlympiadBench, and AMC23, indicating that Cold-start can restrict exploration and transfer poorly to harder problems. While the Naive Alternative partially mitigates this issue, its gains remain limited. In contrast, BRIDGE achieves consistent and substantial improvements on the more challenging benchmarks. These results underscore BRIDGE's superior generalizability in handling complex mathematical reasoning.

| Method | MATH 500 | Minerva Math | Olympiad Bench | AIME24 | AMC23 | Average |
|--------|----------|--------------|----------------|--------|-------|---------|
| Instruct | 38.0 | 14.3 | 13.0 | 13.3 | 25.0 | 20.7 |
| SFT | 38.4 | 10.3 | 11.9 | 27.5 | 3.3 | 18.3 |
| RL-zero | 48.6 | 15.1 | 17.8 | 10.0 | 17.5 | 21.8 |
| Cold-start | 45.0 | 11.8 | 12.0 | 3.3 | 22.5 | 18.9 |
| Naive Alter. | 49.8 | 17.6 | 17.2 | 20.0 | 0.0 | 20.9 (+10.6%) |
| BRIDGE | 51.8 | 15.1 | 19.3 | 10.0 | 27.5 | 24.7 (+30.7%) |

Table 2: Performance on Llama3.2-3B-Instruct.

**Generalization to Out-of-Domain tasks.** To test whether BRIDGE's benefits transfer beyond math into out-of-domain reasoning. We evaluate on two non-math OOD benchmarks, LiveCodeBench and

| Method | MATH 500 | Minerva Math | Olympiad Bench | AIME24 | AMC23 | Average |
|---|---|---|---|---|---|---|
| Base | 55.4 | 24.3 | 22.5 | 3.3 | 27.5 | 26.6 |
| SFT | 67.8 | 32.0 | 29.8 | 45.0 | 13.3 | 37.6 |
| RL-zero | 76.2 | 36.0 | 42.4 | 10.0 | 50.0 | 42.9 |
| Cold-start | 80.4 | 38.2 | 39.6 | 16.6 | 52.5 | 45.5 |
| Naive Alter. | 78.2 | 37.5 | 40.6 | 13.3 | 65.0 | 46.9 (+3.1%) |
| BRIDGE | 79.0 | 39.7 | 44.0 | 16.7 | 70.0 | 49.9 (+9.7%) |

Table 3: Performance on Qwen3-8B-Base.

| Method | LiveCodeBench | GPQA | Avg. |
|---|---|---|---|
| Base | 32.95 | 32.32 | 32.64 |
| Instruct | 33.07 | 35.80 | 34.44 |
| RL-zero | 32.50 | 38.38 | 35.44 |
| Cold-start | 23.86 | 25.76 | 24.81 |
| BRIDGE | 34.55 | 42.93 | 38.74 |

Table 4: OOD evaluation on Qwen3-8B.

| Method | Average Performance | | | Avg. |
|---|---|---|---|---|
| | Epoch=1 | Epoch=3 | Epoch=6 | |
| SFT | 24.1 | 26.5 | 27.9 | 26.2 |
| RL-zero | 14.8 | 17.5 | 32.2 | 21.5 |
| Cold-start | 33.4 | 28.5 | 32.2 | 31.4 |
| Naive Alt. | 13.0 | 30.8 | 33.4 | 25.7 |
| BRIDGE | 32.3 | 33.3 | 36.4 | 34.0 |

Table 5: Performance across training epochs.

GPQA based on Qwne3-8B, as seen in Table 4, BRIDGE attains the best performance, demonstrating that joint training yields consistent cross-domain gains.

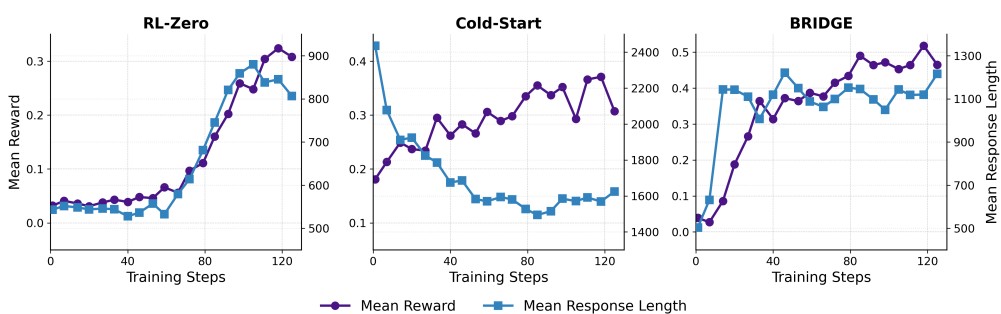

Figure 3: Training dynamics of mean reward and response length on Qwen2.5-3B.

**Training Dynamics Analysis.** We analyze the dynamics of mean reward and response length during training for BRIDGE, Cold-start, and RL-Zero on Qwen2.5-3B. As shown in Figure 3, the three methods exhibit markedly different patterns. RL-Zero suffers from online RL's sample inefficiency, showing slow growth in both response length and reward. Cold-start begins with extremely long responses due to SFT warm-up, *causing slow training* (see in Table 6), followed by a sharp decline and gradual recovery. This "dip-then-rise" pattern indicates the model initially loses expert behavior acquired during SFT, then slowly explores new strategies—a mismatch that contributes to training inefficiency. Despite starting with higher rewards, Cold-start's second-phase RL lacks proper guidance, resulting in final rewards similar to RL-Zero. In contrast, BRIDGE benefits from continuous SFT guidance throughout training, enabling rapid reward growth that surpasses Cold-start and achieving superior convergence. These dynamics demonstrate that BRIDGE's bilevel optimization enables more efficient policy learning through sustained and targeted expert guidance.

Performance on downstream tasks across training epochs shows the same pattern (Table 5): Cold-start learns quickly at the beginning but dips mid-training and finishes no better than RL-zero, indicating constrained exploration. BRIDGE starts strong and improves steadily to the best final performance. Naive Alternating narrows the gap mid-training yet remains below BRIDGE. RL-zero shows the poorest early-stage efficiency. (see Appendix C for details.)

**Cost-Benefit Analysis.** We evaluated the cost-performance trade-offs by measuring wall-clock training time, average GPU memory usage per device, and final convergence performance across two model scales: Qwen2.5-3B (4×A100-80GB) and Qwen3-8B-Base (8×MI300-192GB). As shown in Table 6, Cold-start requires nearly 2x the training time of RL-zero, despite the short SFT stage. This overhead stems from long sequence lengths induced by the SFT stage (see Figure 3). BRIDGE achieved 44% and 14% time savings compared to Cold-start for the 3B and 8B models, respectively. Despite a modest 11% increase in memory usage for the larger model, BRIDGE consistently delivered superior performance improvements (13% for 3B and 9.7% for 8B models), demonstrating favorable cost-benefit trade-offs for practical deployment.

| Metric | Qwen 2.5-3B | | | Qwen 3-8B-Base | | |
|---|---|---|---|---|---|---|
| | RL-zero | Cold-start | BRIDGE | RL-zero | Cold-start | BRIDGE |
| Time (hr) | 6.1 | 12.3 | 6.9 | 38.5 | 39.1 | 33.5 |
| Mem. (GB) | 52.2 | 45.9 | 59.3 | 50.7 | 60.8 | 67.4 |
| Acc. (%) | 32.2 | 32.2 | 36.4 | 42.9 | 45.5 | 49.9 |

Table 6: Cost-performance analysis on Qwen2.5-3B and Qwen3-8B-Base

**LoRA ablation.** We prove BRIDGE is insensitive to LoRA hyperparameters (Appendix D).

## 5 RELATED WORK

SimpleRL (Zeng et al., 2025) observes that fine-tuning on short-CoT datasets can harm reasoning ability, while He et al. (2025) find that fine-tuning on long-CoT distilled data can improve the reasoning performance of smaller models—especially when used as a warm-up stage before RL training. In practice, two-stage pipelines that combine SFT and RL are commonly used to balance stability and performance. However, existing approaches often rely solely on supervised fine-tuning, which tends to generalize poorly, or on pure RL, which suffers from sample inefficiency and unstable optimization.

Recent efforts move beyond the decoupled "SFT then RL" recipe by mixing two objectives within one stage. LUFFY (Yan et al., 2025) combines off-policy expert traces with on-policy rollouts using Mixed-Policy GRPO and regularized importance sampling, but it requires the off-policy demonstrations to be paired with the same prompts as the on-policy data, limiting flexibility and data reuse. CHORD (Zhang et al., 2025) integrates SFT into on-policy RL via a weighted-sum objective with a global weight and a token-level weight that up-weights uncertain expert tokens and down-weights large-divergence tokens. This stabilizes training compared with naïve loss addition, but the coupling is heuristic and offers no mechanism to ensure that the injected supervision is useful for the RL update. BRIDGE instead treats SFT–RL cooperation as a bilevel problem, meta-adapting the supervision to maximize the reward gain of joint training over RL alone and yields larger and more robust improvements than simple loss mixing. It offers a new perspective on integrating imitation and exploration for large reasoning models. We provide an extended discussion of related work in Appendix E.

## 6 CONCLUSION

This work investigates how to effectively integrate supervised fine-tuning and reinforcement learning to improve the reasoning capabilities of LLMs. We begin by analyzing three widely used training paradigms and identify a key limitation of existing multi-stage pipelines: the lack of interaction between SFT and RL. To address this, we propose a simple alternating baseline and further introduce BRIDGE, a bilevel optimization framework that models SFT as the upper-level objective and RL as the lower-level objective. By employing a penalty-based relaxation, BRIDGE explicitly encourages joint training to outperform standalone RL, fostering tighter cooperation between the two learning paradigms. Empirical results on five mathematical reasoning benchmarks demonstrate that our method consistently outperforms strong baselines in both accuracy and training efficiency. These findings underscore the potential of bilevel optimization as a unifying framework for combining supervised and reward-driven learning in complex reasoning tasks.

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

| Rand/$\alpha$ | MATH500 | Minerva | OlympiadBench | AIME24 | AMC23 | Avg. |
|---|---|---|---|---|---|---|
| 32/16 | 79.0 | 39.7 | 44.0 | 16.7 | 70.0 | 49.9 |
| 16/32 | 79.0 | 38.6 | 44.0 | 16.0 | 70.0 | 49.5 |

Table 7: LoRA sensitivity ablation on Qwen3-8B-Base.

## A  THE USE OF LARGE LANGUAGE MODELS

In preparing this manuscript, we used a large language model (LLM) solely for polishing the writing style and improving the clarity of the manuscript. The LLM was not used for generating research ideas, designing experiments, conducting analyses, or deriving results. All scientific contributions, including the conceptualization, methodology, experiments, and conclusions, were developed entirely by the authors.

## B  IMPLEMENTATION DETAILS

All models are trained using the Verl framework (Sheng et al., 2024). We use a prompt batch size of 64, mini-batch size of 64, and learning rate of $5 \times 10^{-7}$. For LoRA, we set both rank and $\alpha$ to 16. The penalty weight $\lambda$ is set to 0.5. We employ two configurations: (1) for 3B models: 5 rollouts per prompt with 3k maximum tokens; (2) for 8B models: 8 rollouts per prompt with 8k maximum tokens. During evaluation, we use greedy decoding (temperature 0) with a 5k or 8k token limit and report pass@1 accuracy. Experiments are conducted on 4×NVIDIA A100 GPUs (80GB) for 3B models and 8×AMD MI300 GPUs (192GB) for 8B models.

## C  PERFORMANCE ON VARIED FINE-TUNING EPOCHS

We assess BRIDGE's effectiveness across different fine-tuning epochs on Qwen2.5-3B using average performance across epochs as the metric. As shown in Table 5, BRIDGE achieves the highest average performance. Among the baselines, Cold-start yields the second-best trade-off. However, its performance becomes unstable as training progresses, eventually converging to the same final result as RL-zero. In contrast, BRIDGE demonstrates consistent improvement throughout training. Overall, nearly all hybrid baselines outperform RL-zero in terms of early-stage efficiency, highlighting the advantage of integrating supervised fine-tuning and reinforcement learning paradigms.

## D  LORA HYPERPARAMETER SENSITIVITY

To ensure BRIDGE's gains do not hinge on a particular LoRA setting and to assess robustness to LoRA. We ablate LoRA rank ($R$) and $\alpha$ on Qwen3-8B-Base, keeping all training settings fixed and only changing $(R, \alpha)$. Table 7 indicates nearly identical outcomes across configurations, confirming that BRIDGE is insensitive to the LoRA choice.

## E  RELATED WORK

**Reinforcement Learning for Large Reasoning Models.**   Recent progress has highlighted the critical role of reinforcement learning in enhancing the reasoning capabilities of large language models (OpenAI; DeepSeek-AI et al., 2025). DeepSeek-R1 introduced a simple yet effective rule-based reward model and demonstrated further gains through multiple rounds of supervised distillation and RL training. LIMR (Li et al., 2025) showed that complex reasoning behaviors can emerge from as few as one thousand curated examples from the MATH dataset (Hendrycks et al., 2021b).

In parallel, substantial advances have been made in training recipes for large reasoning models. Chu et al. (2025) compare SFT and RL for reasoning tasks and find that RL generalizes significantly better, whereas SFT is prone to overfitting. SimpleRL (Zeng et al., 2025) observes that fine-tuning on short-CoT datasets can harm reasoning ability, while He et al. (2025) find that fine-tuning on long-CoT distilled data can improve the reasoning performance of smaller models—especially when

used as a warm-up stage before RL training. In practice, two-stage pipelines that combine SFT and RL are commonly used to balance stability and performance. However, existing approaches often rely solely on supervised fine-tuning, which tends to generalize poorly, or on pure RL, which suffers from sample inefficiency and unstable optimization. In this work, we propose the first unified training framework that enables explicit interaction between SFT and RL via a bilevel optimization formulation. This approach offers a new perspective on integrating imitation and exploration for large reasoning models.

**Bilevel Optimization in LLMs.** Bilevel optimization (BLO) is a classical framework for modeling hierarchical learning problems, originating from Stackelberg leader-follower games. Two major classes of methods have been developed to solve BLO problems. Implicit gradient methods (Hong et al., 2020; Khanduri et al., 2021; Shen & Chen, 2022; Xiao et al., 2023) compute gradients through the lower-level problem using second-order derivatives. While theoretically robust, these methods are often computationally expensive and memory-prohibitive when applied to large-scale models such as LLMs. In contrast, penalty-based relaxation methods (Shen & Chen, 2023; Kwon et al., 2023; Shen et al., 2024; Lu, 2024) approximate the BLO formulation using only first-order gradients, making them substantially more scalable and thus better suited for LLM applications. Recent work has explored the use of bilevel optimization in LLMs for tasks such as data selection (Lin et al., 2024; Shen et al., 2025), inverse reinforcement learning (Li et al., 2024), and meta-learning (Choe et al., 2023; Shirkavand et al., 2025). To the best of our knowledge, our work is the first to cast reasoning-oriented LLM training as bilevel optimization, introducing a novel augmented model architecture for modeling and solving this problem. This provides a principled framework for integrating supervised and reinforcement learning, where SFT actively assists RL optimization rather than merely serving as warmup.

