# OpenReview forum: "Beyond Two-Stage Training: Cooperative SFT and RL for LLM Reasoning"
_ICLR.cc/2026/Conference — ICLR 2026 Conference Withdrawn Submission_

### Official Review · Reviewer_1xj6 · 2025-10-23

**Soundness:** 2
**Presentation:** 2
**Contribution:** 2
**Rating:** 2
**Confidence:** 4

**Summary:**

This paper addresses the inefficiency and sub-optimality of the standard two-stage training paradigm for LLMs in reasoning tasks, where supervised fine-tuning is followed by RL. They introduce a novel framework that formulates the joint training of SFT and RL as a bilevel optimization problem. The method uses an augmented model architecture (base model + LoRA) and a penalty-based relaxation for efficient optimization. Empirical results across three LLMs and several mathematical reasoning benchmarks demonstrate that this method consistently outperforms standard baselines, in both final accuracy and training efficiency.

**Strengths:**

The primary strength of this work lies in its novel formulation of SFT-RL integration. While combining imitation learning and reinforcement learning is not a new concept, framing it as a cooperative meta-learning problem using bilevel optimization is a principled and original approach in the context of LLM reasoning. Moving beyond simple alternating updates or heuristic loss mixing, BRIDGE provides a clear mathematical structure where the SFT objective is explicitly conditioned on the optimal RL policy.

Moreover, it uses basic experiments to empirically validate the efficiency of the proposed method.

**Weaknesses:**

1. Incomplete and misleading information: in the experiment settings, the detailed RL type is not declared and the information for optimizer is missing. And in Equations (2) and (4) $D_{KL}$ is not explained before. Moreover, detailed explaination on LoRA is missing. The architectural choice of splitting parameters into a base model and a LoRA module is claimed to be "essential" to prevent the formulation from collapsing into a MAML-style update. This is a strong claim that is never theoretically explained or empirically verified.

2. Lack of ablation study and the reason for hyper-parameter selection: usually the learning rate for SFT is much larger than that for RL. It is unreasonable to use the same learning rate of 5e-7 for all the experiments. Moreover, this work only implement ablation study on LoRA Rank and $\alpha$, but other ablations, including ablation on $\lambda$ and on learning rate, are not included.

3. Weakened Novelty Claims due to Missing Baselines: The paper positions BRIDGE as a superior, principled alternative to "heuristic" methods that mix SFT and RL objectives. However, it fails to compare against any such modern methods. Recent work like CHORD mentioned in this paper, which uses a dynamic weighted-sum objective, is mentioned but not included as a baseline.

4. The "dip-then-rise" dynamic is presented as a fundamental flaw of two-stage training. However, this issue could potentially be mitigated by simpler methods not explored here. For example, would maintaining a KL-divergence penalty against the SFT-trained model during the RL phase alleviate the forgetting? By not exploring simpler fixes to the existing paradigm, the paper presents BRIDGE as the only solution to a problem that may have less complex remedies.

5. This paper contains a lot of typos. It seems to be an uncomplished or rushed work:
   1. Line 192, Line 265: Eq. equation-> Eq.
   2. Line 202: Equation equation 4->Equation 4.
   3. Line 312: 8,5k; lama-3.2-3B-Instruct->8.5k problems; Llama-3.2-3B-Instruct
   4. Line 312: ...RL training For the SFT... -> ...RL training. For the SFT...
   5. Line 400: Qwne3-8B->Qwen3-8B
   6. Table 7: Rand->Rank

**Questions:**

1. Could you please clarify the discrepancy regarding the penalty weight $\lambda$? How sensitive is BRIDGE's performance to the choice of $\lambda$ and its schedule?

2. The approximation of $\theta^*(w)$ with a single gradient step is a potential weak point. Have you investigated the impact of using more accurate approximations, such as multi-step updates? How does this trade off with computational cost, and does it affect the stability or final performance of the algorithm?

3. Why was a direct comparison to other single-stage SFT+RL methods like CHORD omitted?

---

### Official Review · Reviewer_fa3w · 2025-10-28

**Soundness:** 3
**Presentation:** 2
**Contribution:** 3
**Rating:** 4
**Confidence:** 2

**Summary:**

This paper aims at fusing the two stages training, SFT followed by RL, into one single stage. The paper first argues that RL after SFT has several drawbacks, namely forgetting and inefficient exploration. To mitigate this issue, the paper propose BRIDGE, which split the parameter into "base model" and LoRA, where the LoRA portion maximize the SFT objective, given the base model maximizing the RL objective. Experiments several models and datasets shows that proposed method outperforms selected baselines.

**Strengths:**

The strengths of the paper are listed as follows

1. This paper propose an alternative to the traditional SFT-RL training framwork. The paper argues several drawback of these framework and propose an interesting synergy to mitigate this.

2. The training are conducted on three models and evaluation takes on five datasets, and BRIDGE demonstrates good performances, without introducing excessive extra computation.

3. The ablation study in Figure 3 provides a good illustration to the motivation of the proposed method,

**Weaknesses:**

Some weaknesses of this paper is listed as follows

1. The design of the architecture, namely equation (4), looks somewhat arbitrary. My understanding is that (4) aims at finding a pair of ($\theta, w$) such that they serve as the optimal if another is fixed. While this makes sense to me, I don't see a specific reason why assigning LoRA parameter for SFT objective and base model portion for RL objective, and RL objective serves as the constraint here. Could the authors provide more rationale on this?

2. Parts of the paper are not clear enough

    a. In the caption of Figure 1, it would be better if a brief explanation of the three methods

    b. At line 141, what is the dataset of grade 3-5 level math?

    c. Line 251, what are the required regularity condition? How reasonable it it to assume $J_{RL}$ satisfying this condition?

    d. What RL algorithm is used in the experiment (e.g., PPO, GRPO, etc)? How does different RL algorithm affect the experiment results?

**Questions:**

See weakness section.

---

### Official Review · Reviewer_qWx6 · 2025-10-30

**Soundness:** 2
**Presentation:** 3
**Contribution:** 1
**Rating:** 2
**Confidence:** 3

**Summary:**

This paper addresses inefficiencies in the standard two-stage "SFT-then-RL" training pipeline for LLM reasoning. The authors identify that this "cold-start" approach suffers from a distribution mismatch, leading to catastrophic forgetting and a "dip-then-rise" training dynamic. They propose BRIDGE, a novel framework that reformulates the joint training as a bilevel optimization (BLO) problem. In this formulation, SFT acts as the upper-level problem, optimizing LoRA parameters to "meta-learn" how to guide the lower-level RL problem, which optimizes the base model parameters. The BLO is solved efficiently using a first-order, penalty-based relaxation. Experiments across three LLMs and five reasoning benchmarks show that BRIDGE consistently outperforms SFT, RL-only, and the standard cold-start baselines in both final accuracy and training efficiency compared to the cold-start method.

**Strengths:**

- The formulation of SFT as the upper-level "leader" and RL as the lower-level "follower" is highly intuitive. This perspective effectively frames SFT as a meta-learning process designed to provide targeted guidance for the RL stage.
- The intuition behind the update rules is clearly explained in Section 3.3. The "cooperative gain" term, in particular, provides a clear mechanism for optimizing the LoRA parameters ($w$) to maximize the advantage of joint training over an RL-only approach.
- The authors validate their method, BRIDGE, across three distinct LLMs (Qwen2.5-3B, Llama-3.2-3B-Instruct, and Qwen3-8B-Base). This effectively demonstrates the method's robustness across different model families and scales.

**Weaknesses:**

- The paper suffers from a significant omission of key related work that also explores the integration of SFT and RL. Consequently, the experimental baselines are not comprehensive and fail to include comparisons against state-of-the-art algorithms from this body of work. This omission makes it difficult to assess the true performance and contribution of BRIDGE. Relevant missing works include, but are not limited to:
    * [1] UFT: Unifying Supervised and Reinforcement Fine-Tuning
    * [2] SRFT: A Single-Stage Method with Supervised and Reinforcement Fine-Tuning for Reasoning
    * [3] Learning What Reinforcement Learning Can't: Interleaved Online Fine-Tuning for Hardest Questions
    * [4] On-policy rl meets off-policy experts: Harmonizing supervised fine-tuning and reinforcement learning via dynamic weighting

- A central methodological question is the justification for the complex Bi-Level Optimization (BLO) framework.
    - **Derivation vs. Heuristic:** The final update rule for the base parameters $\theta$ (Eq. 8) surprisingly simplifies to a straightforward curriculum-weighted average of SFT and RL gradients. The paper does not convincingly argue why this complex BLO derivation is necessary to arrive at an update rule that could, arguably, be motivated heuristically as a simple curriculum learning strategy. Does the BLO formulation provide demonstrable value beyond a simpler, heuristically-defined weighting schedule?
    - **Algorithmic Complexity:** Relatedly, the paper does not sufficiently justify the leap in complexity from the simpler alternating update schedule (Algorithm 1) to the full BLO machinery (Algorithm 2), especially when Algorithm 1 already achieves a large portion of the gains. The marginal improvement observed does not seem to warrant this significant increase in algorithmic complexity.

- A significant concern is the practical overhead of the method. The cost-benefit analysis in Table 6 clearly shows that BRIDGE incurs the highest GPU memory overhead of all compared methods (e.g., 59.3 GB on the 3B model, versus 52.2 GB for RL-Zero and 45.9 GB for Cold-start). This high cost may limit its practical applicability.

- The abstract claims that "early RL forgets SFT-acquired behavior and slowly re-explores." This serves as a key motivation for the proposed method. However, the paper does not appear to provide empirical evidence to substantiate this claim.

**Questions:**

Please see **weaknesses**.

---

### Official Review · Reviewer_T7vy · 2025-10-31

**Soundness:** 2
**Presentation:** 2
**Contribution:** 2
**Rating:** 4
**Confidence:** 4

**Summary:**

BRIDGE is a bilevel co-training scheme for SFT+RL. Base params $\theta$ get a $\lambda$-weighted mix of SFT/RL gradients; a small LoRA branch w is updated to maximize a “cooperative gain” surrogate (how much joint training beats RL-only after a one-step RL proxy).

BRIDGE is intended to avoid two-stage brittleness (SFT -> RL) and reduce early RL instability/catastrophic forgetting by explicitly shaping the SFT-side signal during RL.

Results (as reported): On Qwen2.5-3B, Llama-3.2-3B-Instruct, and Qwen3-8B, BRIDGE often improves average pass@1 over SFT, RL-Zero, Cold-Start, and a Naïve Alternating baseline across five math benchmarks; some OOD results (LiveCodeBench/GPQA) are shown for 8B.

**Strengths:**

1. The paper frames cooperation as a bilevel problem (SFT as upper level, RL as lower level) and then relaxes it into a workable first-order update: $\theta$ gets a $\lambda$-weighted mix of SFT/RL gradients and the LoRA $w$ is updated to maximize a ``cooperative gain.'' The objective, penalty, and Algorithm 2 are spelled out and easy to reimplement.
2. They compare BRIDGE to SFT, RL-zero, Cold-start, and a na\"{i}ve alternating baseline, and report average gains across five math benchmarks on Qwen2.5-3B, Llama-3.2-3B-Instruct, and Qwen3-8B (e.g., +11.8\% vs.\ Cold-Start on one setup). They also include LoRA sensitivity showing robustness to adapter hyperparameters. Implementation details (rollouts, token caps, eval settings) are documented.
3. Generalization results on LiveCodeBench are interesting.

**Weaknesses:**

1.  The catastrophic forgetting argument is not backed by strong evidence. It could be potentially an artifact of the 3k token limit during the RL training phase. Hence I find figure 3 misleading. Did the authors try SFT with a max token budget of say 1500 and then perform rl? This would drastically change the claims across the paper I feel including the computational cost claims I feel.
2. The relative improvements in the results can be misleading if not read carefully. It would be preferable to have absolute improvement numbers.
3. Table 4 (OOD: LiveCodeBench, GPQA) reports Base/Instruct/RL-zero/Cold-start/BRIDGE, but omits naive alternating baseline. That weakens the claim that BRIDGE is best beyond math---its closest in-domain competitor is not shown OOD.
4.  They argue the LoRA split is essential (otherwise it collapses to MAML and RL learning is disabled), but there is no ablation with full-parameter upper level or alternate adapters. The only LoRA ablation tweaks rank/$\alpha$ (Table 7), which is narrow. Architectural necessity is asserted, not demonstrated.
5. (Minor but relevant) Method text says $\lambda$ is annealed, yet the appendix fixes $\lambda=0.5$. Clarify and add a $\lambda$-sensitivity.

**Questions:**

1. The evaluation setup is not clear in figure 1. What is the base model used here? What is the dataset for training and what is the evaluation dataset?
2. The “Base” in Table 1 corresponds to the qwen2.5 3B pretrained model?
3. How are the SFT and Base evaluations done in Table 1, 2, etc? Because I am looking at the Qwen 2.5 technical report (Table 5 for example) and the reported MATH score for 3B qwen model is 42.6 whereas the reported SFT score in Table 1 is 53.4 and base score 32.4. Why is there such a huge discrepancy?

---

### Note · Authors · 2025-12-31

I have read and agree with the venue's withdrawal policy on behalf of myself and my co-authors.